

# Different roles of Ca²⁺ and chitohexose in peanut (*Arachis Hypogaea*) photosynthetic responses to PAMP-immunity

Quan Wang[1,*], Ye Zhang[2,*], Li Cui[1], Jingjing Meng[1], Sha Yang[1], Xinguo Li[1] and Shubo Wan[3]

[1] Institute of Crop Germplasm Resources, Shandong Academy of Agricultural Sciences, Ji'nan, China
[2] HuangShan University, College of Life and Environment Sciences, Huangshan, China
[3] Shandong Academy of Agricultural Sciences, Ji'nan, China
* These authors contributed equally to this work.

## ABSTRACT

**Background:** During active infections, plants prevent further spread of pathogenic microorganisms by inducing the rapid programmed death of cells around the infection point. This phenomenon is called the hypersensitive response and is a common feature of plant immune responses. Plants recognize conserved structures of pathogenic microorganisms, called pathogen-associated molecular patterns (PAMPs), *e.g.*, flagellin 22 (flg22) and chitohexose, which bind to receptors on plant cells to induce various immune-response pathways. Although abiotic stresses are known to alter photosynthesis, the different effects of flg22 and chitohexose, which are involved into PAMP-induced signaling, on photosynthesis needs further study.
**Methods:** In the present study, we assessed the role of PAMPs in peanut (*Arachis hypogaea*) photosynthesis, particularly, the interaction between PAMPs and Ca²⁺ signal transduction pathway.
**Results:** Both flg22 and chitohexose significantly promoted the expression of the pathogenesis-related genes *PR-4* and *PR-10*, as did Ca²⁺. We found that Ca²⁺ is involved in downregulating the photosystem II (PSII) reaction center activity induced by the flg22 immune response, but the role of chitohexose is not obvious. Additionally, Ca²⁺ significantly reduced the non-photochemical energy dissipation in the flg22- and chitohexose-induced immune response.
**Conclusion:** These results indicated that flg22 and chitohexose can trigger peanut immune pathways through the Ca²⁺ signaling pathway, but they differ in their regulation of the activity of the PSII reaction center.

## INTRODUCTION

Peanut (*Arachis Hypogaea*) is an important oil crop in China, and throughout the world. In addition to uses as oil and food, peanut also provides raw material for poultry and aquatic feed processing because the byproducts of peanut processing are rich in protein. In its late growth period, peanut leaves often suffer from disease, which seriously affects the yield and quality of the resulting peanut commodity.

Corresponding authors
Sha Yang, yangsha0904@126.com
Xinguo Li, xinguol@163.com

In natural environments, plants are inevitably exposed to microorganisms, including pathogens that infect plants to achieve their own growth and reproduction. When plants are infected, the aging and shedding of their leaves can be accelerated, which can ultimately affect crop yields. Rapid programmed death of plant cells around the infection site helps prevent the further spread of pathogenic microorganisms. This hypersensitivity is a common mechanism of plant immune responses. Plants recognize the essential components of microbial conservation through pattern recognition receptors (PRR) to quickly start the immune pathway (*Lu et al., 2010*). The pathogenic microbial conservation components that can be recognized by this receptor are called pathogen-associated molecular patterns (PAMP), and then release a variety of signaling molecules, such as calcium ions ($Ca^{2+}$), reactive oxygen species (ROS) and various plant hormones. This mechanism evolved through the long-term interaction of pathogens and their host plants, and the PAMP-triggered immune response is key to understanding plant immunity (*Jie et al., 2007*).

Plants use light energy to convert carbon dioxide ($CO_2$) and $H_2O$ into organic matter in chloroplasts and release oxygen, which is known as photosynthesis. Photosynthesis is divided into the light reactions and the dark reactions. The light reactions mainly involve photosystem II (PSII) and photosystem I (PSI) proteins, which convert light energy into chemical energy, while the dark reaction uses the energy and substances generated in the light reaction to convert $CO_2$ into organic substances, which is called the Calvin cycle. Plant photosynthesis is responsible for most of the production of oxygen and the fixation of biomass on Earth (*Hankamer, Barber & Boekema, 1997*). Therefore, when plants are stressed, their cells will devote a lot of energy to resist stress. In this case, the physiological and biochemical indexes of plants will inevitably decrease, including those of photosynthesis.

The flagellum is the motor organ of bacteria, enabling these single-celled organisms to move in response to stimuli (*Hajam et al., 2017*). Flagellin is derived from the conserved N-terminal or C-terminal of the flagella of various bacteria. Animal and plant cells can recognize flagellin and respond to bacterial infections before they have a chance to take hold. In 1999, *Felix et al. (1999)* purified flagellin from *Pseudomonas syringae* and synthesized a highly conserved amino acid residue sequence at the N-terminal as a stimulus to treat plant cells. They found that plant cells could rapidly produce ROS and other substances in response to this flagellin treatment, and identified flg22 as a key stimulus factor for some plants to recognize bacteria and induce an immune response (*Felix et al., 1999*). *Gómez & Boller (2000)* screened Arabidopsis (*Arabidopsis thaliana*) mutants treated with flg22 and identified the receptor FLAGELLIN-SENSITIVE 2 (FLS2) as being involved in recognizing flagellin. FLS2 has a transmembrane domain, with flg22 binding to its extracellular domain (*Dunning et al., 2007*). In addition to flg22, chitosan has been widely used in agriculture to increase the ability of plants to resist stress. Although some studies have found that chitosan can induce plant cells to produce a defense response, and even programmed cell death, more studies believe that chitosan can stimulate a series of defense responses and enhance plant resistance to stress (*Katiyar, Hemantaranjan & Singh, 2015*; *Jia et al., 2019*; *Khan, Prithiviraj & Smith, 2003*).

Environmental signals trigger rapid and transient increases in cytosolic $Ca^{2+}$ in plants, which in turn activates signal transduction pathways involved in many physiological and biochemical processes, particularly the responses to abiotic and biotic stresses in plants (*Bowler & Fluhr, 2000*; *Schreiber, Bilger & Neubauer, 1994*). Our previous studies and those of others have shown that the $Ca^{2+}$ signaling pathway plays a regulatory role in photosynthesis, influencing the turnover of PSII reactive center protein components and the non-photochemical quenching of chlorophyll fluorescence (NPQ) (*Brunner, 2002*; *Yang et al., 2013*). NPQ might be involved into response of PAPM-triggered immunity, and the regulation of NPQ might be an intrinsic component of the plant's defense program with flg22 treatments (*Göhre et al., 2012*).

There are many common signaling molecules in the immune and photosynthetic pathways of plants, such as $Ca^{2+}$ and ROS. In this study, besides flg22, we also used chitohexose, another highly effective plant defense stimulator derived from fungal pathogens, to study the induction of the immune response and photosynthesis in peanut leaves, especially their different function mechanisms in the immune response and photosynthesis induction, with an additional analysis of the role of $Ca^{2+}$ signal transduction pathway on the photosynthesis and PAMPs.

## MATERIALS AND METHODS

### Plant materials, growth conditions and treatments

Huayu 25, a peanut (*Arachis Hypogaea*) cultivar variety, was used as material in this study, which was cultured in small plastic pots containing quartz sand. The upper diameter of the pot is 9 cm, the lower diameter is 6.5 cm and the height is 8 cm. The seedlings incubated without $Ca^{2+}$ were marked as NC, and those incubated with Hoagland nutrient solution were marked as CA. The concentration of calcium nitrate tetrahydrate used in the Hoagland nutrient solution is 945 mg/L. The plants were grown at 25/20 °C (day/night) under a 14 h photoperiod (300 µmol $m^{-2}$ $s^{-1}$ photon flux density (PFD)) for 20 d in a greenhouse. Calcium nitrate tetrahydrate was completely removed from Hoagland nutrient solution and balanced nitrogen with ammonium bicarbonate in NC. After about 20 days of cultivation, fifteen peanut functional leaves were taken for three replicates each treatment and placed in a petri dish with deionized water for dark adaptation 4 h. The initial data of CA and NC groups were measured, and then incubated with 1 µM flg22 and 200 g $ml^{-1}$ chitohexose at 25 °C in the dark for 1, 2 and 4 h, respectively.

### Chloroplast fluorescence measurement

Determination of chlorophyll fluorescence with a portable fluorometer (FMS2, Hansatech, Norfolk, UK) according to the protocol described (*Zargar, Asghari & Dashti, 2015*). A Handy plant efficiency analyser (PEA) was used to measure photosynthetic parameters after dark adaptation. Photosynthetic parameters were calculated with reference to Strasser (*Zargar, Asghari & Dashti, 2015*). After fully dark adaptation of isolated leaves in each group, $PI_{(abs)}$, a performance index based on absorbed light energy, was measured in the whole process by a continuous excitation fluorescence analyzer (Handy PEA; Nu-Tech International, New Delhi, India), which was attached to the leaf

clip on the probe. NPQ was estimated as NPQ = Fm/Fm'$^{-1}$ according to *Yang et al. (2015)*, where Fm was measured after dark adaptation for more than 2 h at room temperature prior to stress, Fm' is the maximum intensity of fluorescence in light-acclimated leaves.

## Determination of ROS

Hydrogen peroxide ($H_2O_2$) concentration was measured according to the method of *Sairam & Srivastava (2002)* with modifications. The assay for superoxide anion ($O_2^{\bullet-}$) was performed as described in *Wang & Luo (1990)*.

## The total RNA extraction and real time PCR

Total RNA was extracted from the peanut leaves with the RNA simple kits (Tiangen Biotech, Beijing, China) according to the manufacture's protocol. cDNAs were reverse transcribed using the PrimeScript$^{TM}$ first-strand cDNA synthesis kit (K1622; Thermo scientific, Waltham, MA, US). The polymerase chain reaction (PCR) was amplified following the instruction of SYBR *Premix Ex Taq*$^{TM}$ (TaKaRa, Inc., Dalian, China) with the qRT-PCR amplification instrument (ABI 7500; Applied Biosystems, Waltham, MA, USA). The *TUA5* gene was used as control to calculate the relative expression level.

The sequences of *PR-4*, *PR-10*, *VDE*, *CP12* genes and photosystem b (Psb) family gene *PsbS* were obtained from NCBI. The accession number of the NCBI for *PR-4*, *PR-10*, *PsbS*, *VDE* and *CP12* genes were XM_025821199.2, DQ813661, XM_025812746, XM_025807336 and NC_037358 respectively. Primer sequences were as follows: *TUA5*-F (5′-3′): CTGATGTCGCTGTGCTCTTGG; *TUA5* –R (5′-3′): CTGTTGAGGTTGGTG TAGGTAGG; *PR-4* F (5′-3′): TGGATACAAGAAGGGTCAC; *PR-4* R (5′-3′): GTTG TCCTTTCGAGATAA; *PR-10* F (5′-3′): ATGGGCGTCTTCACTTTCG; *PR-10* R (5′-3′): TGAGTTTCTTGATGGTTCC; *PsbS* F (5′-3′): TTGTTGGTCGTGTTGCCATGATTG; *PsbS* R (5′-3′): ACGGTCACCAAGTGCTCCAATG; VDE, F (5′-3′): TCAGTTGATGCTG TTGACGCTCTC; R (5′-3′): GCAACATTGGCTGCACATGATGG; *CP12* F (5′-3′): AGG AGGCCGAGGAAGCATGTAC; R (5′-3′): CGCTCAGCTCCTCTACCTCATCC.

## Statistical analysis

Statistical significance between groups was evaluated with one-way analysis of variance (ANOVA) followed by Duncan's multiple range test in SPSS Statistics 20.0 (SPSS Inc., Chicago, IL, USA). The mean ± standard error was calculated from three biological replicates per treatment group for each assay. Differences were considered statistically significantly at $p < 0.05$.

# RESULTS

## Flg22 and chitohexose induce pathogenesis-related expression

PAMPs or pathogenic microorganisms can trigger immune responses in plants. For example, both ROS signaling and mitogen-activated protein kinase (MAPK) cascades are activated upon flg22 treatments, which regulate the expression levels of the pathogenesis-related (PR) genes (*Tornero et al., 1997*). In the present study, *PR-4* and *PR-10* were selected to assess the effects of flg22 and chitohexose in peanut leaves. *PR-4* is

associated with chitinase and has antifungal activity, whereas the *PR-10* gene family encodes small cytoplasmic proteins with roles in RNA enzymes and post-translational modifications, and which respond to both biotic and abiotic factors (*Kaku et al., 2006*). Chitohexose belongs to chitohexose fragments, and chitohexose oligosaccharide is a highly effective plant defense stimulator. Both flg22 and chitohexose significantly promoted the expression of *PR-4* (Figs. 1A and 1C) and *PR-10* genes (Figs. 1B and 1C), and this increase in expression was more evident in the presence of $Ca^{2+}$ (Fig. 1), reaching up to a 12-fold difference between the NC and CA expression levels. These results suggest that peanut responds to plant immune regulation positively, and that the $Ca^{2+}$ signal transduction pathway is involved in the immune response caused by flg22 and chitohexose.

## Effects of flg22 and chitohexose on the activity of the PSII reaction center

$PI_{(abs)}$ is an important parameter that can comprehensively reflect the density, absorption and electron transfer of the PSII optical reaction center. When treated with flg22, the $PI_{(abs)}$ of the CA group decreased and then remained unchanged, while that of the NC group increased slightly and then remained stable (Table 1). Compared with the untreated samples, the $PI_{(abs)}$ of the CA group was lower while that of the NC group was higher (Table 1). The $PI_{(abs)}$ of the NC group did not change significantly when treated with chitohexose, while the $PI_{(abs)}$ of the CA group decreased at first but almost returned to the initial level after 4 h of treatment (Table 1). These findings indicate that the immune response triggered by flg22 reduces the number of reaction centers and the performance index of leaf light absorption. The decreased number of reaction centers would also reduce of the maximum light conversion efficiency and photosynthetic performance index. Since $Ca^{2+}$ has a positive response to flg22 but the effect of chitohexose treatment on the PSII reaction center of peanut leaves seems to have little relationship with $Ca^{2+}$ signal transduction pathway.

## Flg22 and chitohexose induce *CP12* and *PsbS* expression

*PsbS* encodes one of the PSII protein components related to NPQ. Under the flg22 treatment, the expression of *PsbS* gene was upregulated in the NC group, but downregulated trend in the CA group, other than a brief increase for the first hour of the treatment; its expression level in the CA group was significantly lower than in the NC group (Fig. 2A). When treated with chitohexose, *PsbS* was downregulated in both the CA group and the NC groups, with no obvious difference between them (Fig. 2C).

Chloroplast protein 12 (CP12) participates in the Calvin cycle, which was localized in the chloroplasts. When treated with flg22, *CP12* was downregulated in the CA group, although its expression slowly increased at the end of the treatment without reaching the initial expression level. This gene was also upregulated in the NC group, and its expression level was higher than in the CA group (Fig. 2B). When treated with chitohexose, *CP12* was downregulated in both the NC and CA groups, although it was upregulated for the first 2 h in the CA group. Throughout the treatment, the *CP12* expression level was higher in the CA group than that in the NC group (Fig. 2D). These results suggest that flg22 induced the

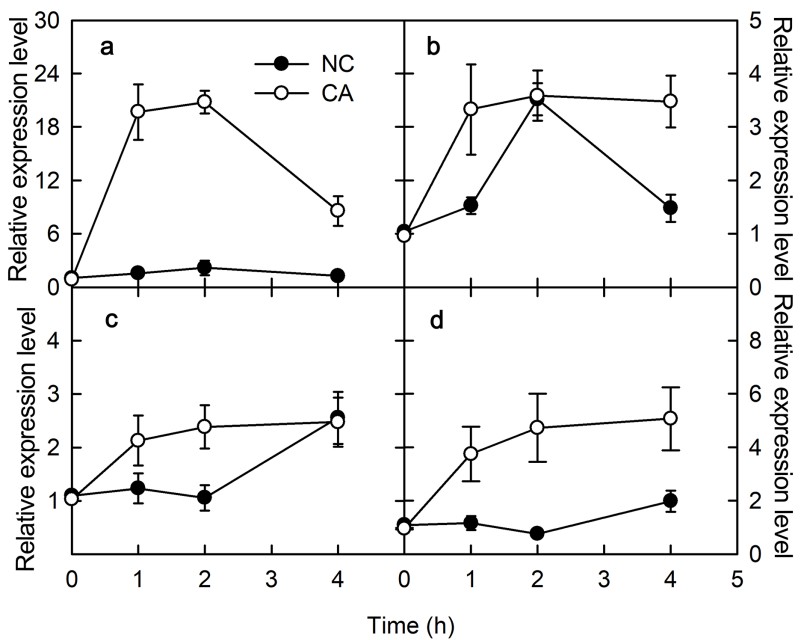

**Figure 1 Effects of FLG22 and chitohexose treatments on expression level of pathogen related genes in peanut leaves.** (A) Expression of *PR-4* gene in peanut leaves treated with 1 µM FLG22; (B) Expression of *PR-10* gene in peanut leaves treated with 1 µM FLG22; (C) Expression of *PR-4* gene in peanut leaves treated with 200 µg/ml chitohexose; (D) Expression of *PR-10* gene in peanut leaves treated with 200 µg/ml chitohexose. Means ± SD.

expression of *CP12* while chitohexose reduced it, and Ca²⁺ inhibited these effects, particularly for the flg22 treatment.

## Influence of the flg22- or chitohexose-triggered immune response on energy dissipation

The plant immune response is more efficient when photosynthesis is inhibited *via* a MAPK cascade signal. The captured light energy is mainly consumed through three pathways during photosynthesis: photochemical electron transfer, chlorophyll fluorescence emission, and heat dissipation during photosynthesis. Among them, photochemical electron transport is related to the biosynthesis of photosynthates, and chlorophyll fluorescence emission consumes little light energy, with any excess energy causing damage to PSII through the photooxidation induced by the accumulation of ROS. Therefore, heat dissipation was an important way to consume excess light energy and prevent photodamage. To excess excitation energy as heat dissipation in a harmless way was called NPQ. Violaxanthin de-epoxidase (*VDE*) encodes Violaxanthin de-epoxidase, which interacted with *PsbS* to participate the regulation of NPQ. *VDE* was downregulated for 2 h after a treatment with flg22, and slightly upregulated after 4 h in both the NC and CA groups. The expression of *VDE* was similar in the CA and NC groups, although at 4 h it was significantly lower in the CA group compared with NC group (Fig. 3A). Meanwhile, *VDE* was upregulated after 1 and 2 h of treatment but downregulated at 4 h of treatment with chitohexose and Ca²⁺, while it was upregulated at 1 h but downregulated at both 2 and

**Table 1 Effects of FLG22 and chitohexose treatments on the activity of photosystem reaction center indicated by $PI_{(abs)}$.**

| Time (h) | FLG 22 (1 μM) | | Chitohexose (200 μg/ml) | |
|---|---|---|---|---|
| | NC (Means ± SD) | CA (Means ± SD) | NC (Means ± SD) | CA (Means ± SD) |
| 1 | 8.7832 ± 0.2625 | 9.1767 ± 0.372 | 9.5630 ± 1.02834 | 10.0273 ± 1.0144 |
| 2 | 9.2564 ± 0.02548 | 7.8374 ± 0.3453 | 9.6554 ± 1.16634 | 9.6740 ± 1.53454 |
| 3 | 9.0753 ± 0.3731 | 7.9554 ± 0.3327 | 9.8453 ± 0.82384 | 9.0673 ± 0.74309 |
| 4 | 8.8453 ± 0.5834 | 7.9443 ± 0.1899 | 9.8026 ± 0.5864 | 9.8632 ± 1.18847 |

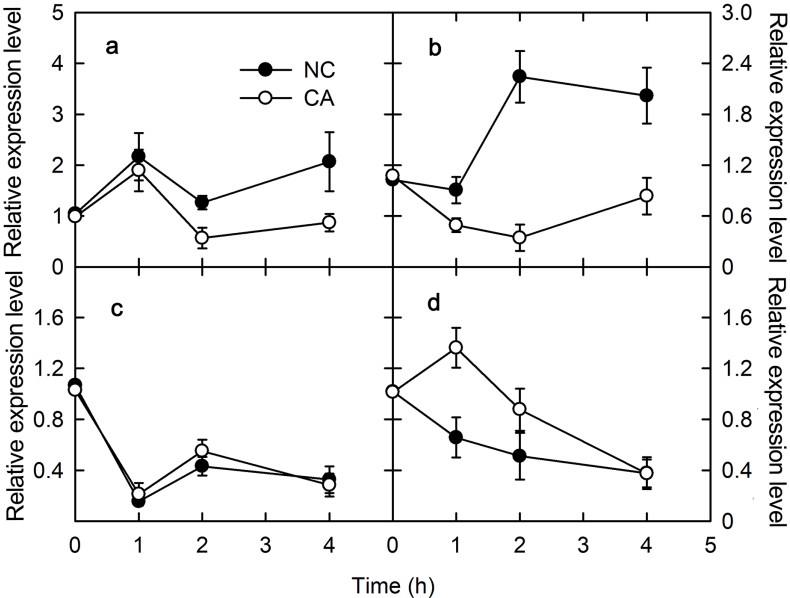

**Figure 2 Effects of FLG22 and chitohexose treatments on expression level of photosynthesis-related genes *CP12* and *PsbS* in peanut leaves.** (A) Expression of *PR-4* gene in peanut leaves treated with 1 μM FLG22; (B) expression of *PR-10* gene in peanut leaves treated with 1 μM FLG22; (C) expression of *PR-4* gene in peanut leaves treated with 200 μg/ml chitohexose; (D) expression of *PR-10* gene in peanut leaves treated with 200 μg/ml chitohexose. Means ± SD.

4 h in the NC group (Fig. 3C). These results suggest that, unlike flg22, chitohexose could induce *VDE*, while $Ca^{2+}$ promoted the expression of *VDE*.

NPQ can reflect excess energy dissipation in plants and is related to the PsbS protein and the xanthophyll cycle. As shown in Fig. 3, flg22 significantly decreased NPQ both in the presence and absence of the $Ca^{2+}$ treatment, with a greater NPQ in the NC group. When treated with chitohexose, the NPQ of both the CA and NC groups showed a downward trend, with the former being significantly lower than the latter (Figs. 3B and 3D). These results suggest that both the flg22- and chitohexose-induced immune responses can reduce the non-photochemical energy dissipation in peanut leaves, and $Ca^{2+}$ is involved in this decrease in both cases.

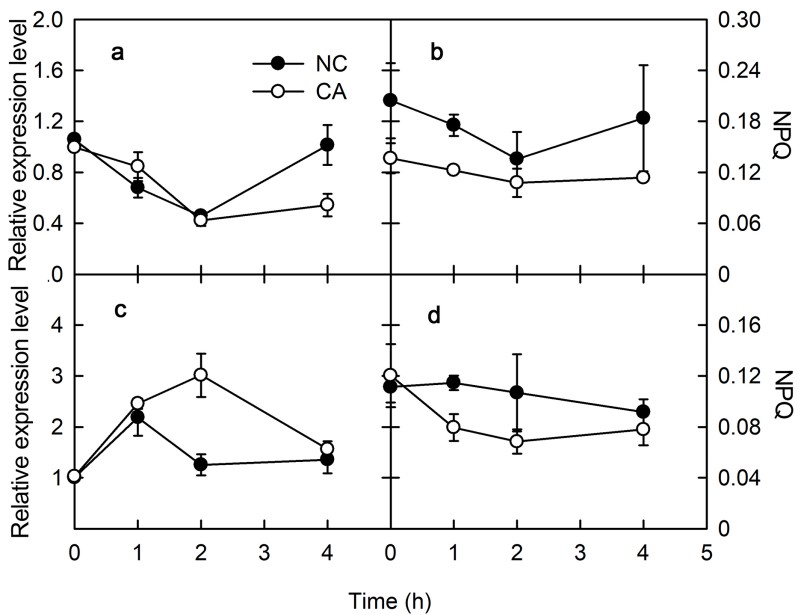

**Figure 3 Effects of FLG22 and chitohexose treatments on expression level *VDE* gene and NPQ in peanut leaves.** (A) Expression of *VDE* gene in peanut leaves treated with 1 µM FLG22; (B) NPQ in peanut leaves treated with 1 µM FLG22; (C) Expression of *VDE* gene in peanut leaves treated with 200 µg/ml chitohexose; (D) NPQ in peanut leaves treated with 200 µg/ml chitohexose. Means ± SD.

## Chitohexose and flg22 alter the ROS content

ROS play important roles in various physiological processes in plants, such as the immune response, development, cell elongation, and phytohormone signaling (*Wan, Zhang & Stacey, 2008*). When exposure to $Ca^{2+}$, plant immune regulation is often accompanied by the rise of ROS, which act as signaling molecules in the immune pathway and enhance the ability of cells to resist pathogenic microorganisms (*Bautista-Baños & Hernández-López, 2004*). The rise of $H_2O_2$ and $O^{2.-}$ occurs in the early defense response following PAMP recognition by plants (*Li, Linhardt & Cao, 2016*). There was no obvious difference in the $H_2O_2$ content between the NC group and the CA group under either the flg22 or the chitohexose treatment (Figs. 4A and 4C). The $O_2^-$ content in the cytoplasm of the peanut leaves began to increase during the first 1 h of both the flg22 and chitohexose treatments, with the NC groups showing slightly higher levels than the CA groups after 3 h (Figs. 4B and 4D), which may be related to the $Ca^{2+}$ signaling pathway improving the ROS-scavenging capacity of the plants (*Katiyar, Hemantaranjan & Singh, 2015*). These results indicated that the rise of ROS is an important part of the peanut immune response, but that $Ca^{2+}$ only has a slight effect on the accumulation of $O_2^-$ (Figs. 4B and 4D).

## DISCUSSION

Peanut (*Arachis Hypogaea*) is an oil crop that supports food security and economic development. When leaves are invaded by pathogens, plants devote a lot of energy to resisting the threat. In this process, photosynthesis and other physiological and biochemical indicators will inevitably be reduced. The main purpose of this study was to

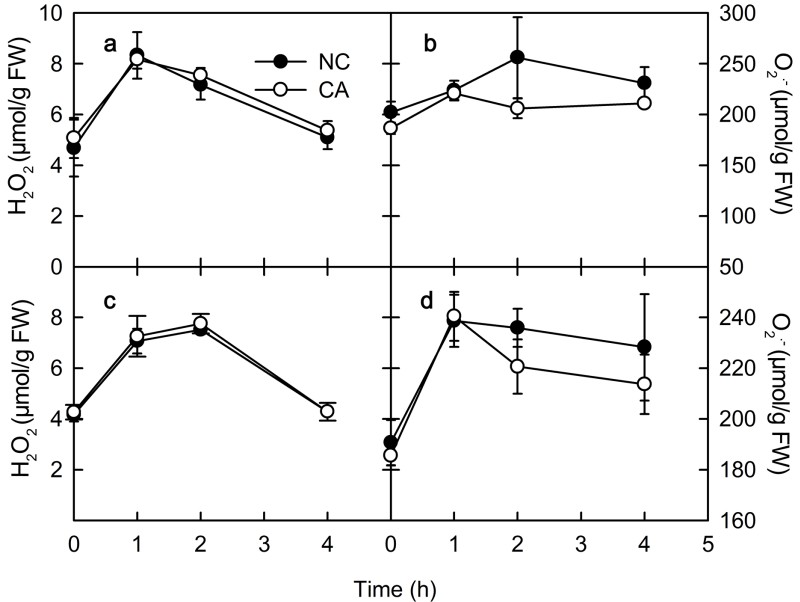

**Figure 4 Effects of FLG22 and chitohexose treatments on content of $H_2O_2$ and $O_2^-$ in peanut leaves.**
(A) Changes of content of $H_2O_2$ in peanut leaves treated with 1 μM FLG22; (B) Changes of content of $O_2^-$ in peanut leaves treated with 1 μM FLG22; (C) Changes of content of $H_2O_2$ in peanut leaves treated with 200 μg/ml chitohexose; (D) Changes of content of $O_2^-$ in peanut leaves treated with 200 μg/ml chitohexose. Means ± SD.                                

investigate the effects of flg22 and chitohexose on the immune response and photosynthesis in peanut. $Ca^{2+}$ is an essential element in plant growth and plays a regulatory role in plant photosynthesis (*Yang et al., 2015*; *Yang et al., 2013*). Therefore, we analyzed the synergistic effect of $Ca^{2+}$ in the flg22- and chitohexose-induced immune response and regulation of photosynthesis.

Flagellin is a conserved protein derived from the flagella of various bacteria, and is a PAMP recognized by plant cells. Chitohexose is a natural biodegradable polymer material, which is widely used in beauty, food, biology, medicine, agriculture, and other fields (*Bautista-Baños & Hernández-López, 2004*). Chitohexose, a fragment of chitosan belonging to saccharide PAMPs, has the same biological effects, although it is a high polymer (*Li, Linhardt & Cao, 2016*).

*PR-4* and *PR-10* belong to the *PR* gene family, and are important components of the plant immune system that can be induced by biotic or abiotic stress (*Liu et al., 2014*; *Du et al., 2017*). The expression of both *PR-4* and *PR-10* increased following the flg22 and chitohexose treatments (Fig. 1), which was accompanied by a high ROS content in the peanut leaves (Fig. 4), suggesting that flg22 and chitohexose trigger an immune response to improve the plant's ability to cope with environmental stress. Plants often respond to stress by forming ROS (*Vera-Jimenez & Nielsen, 2013*), which enhance their stress tolerance, promote tissue repair, and increase pathogen resistance (*Asada, 2006*). As a secondary messenger, $Ca^{2+}$ participates in the regulation of most of the physiological metabolic processes in the cell. Many experimental studies have shown that $Ca^{2+}$ signaling is also involved in the signal transduction process of plant–pathogen interactions (*Ma et al.,*

*2009*). Here, we showed that $Ca^{2+}$ inhibits the accumulation of $O_2^-$ in plant immune processes during the response to flg22 and chitohexose (Figs. 4B and 4D). In addition, $Ca^{2+}$ synergistically induces the expression of *PR-4* and *PR-10* in peanut leaves, enhancing the regulation of the pathogen-related genes by flg22 and chitohexose in peanut leaves (Fig. 1).

When the plant defense signal reaches the cytoplasm, it is further transmitted to each organelle. There are $Ca^{2+}$-related channels in the chloroplast membrane that transmit signals to the chloroplast bodies. The performance index $PI_{(abs)}$ based on the absorption of light energy showed that $Ca^{2+}$ is involved in the activity of the PSII reaction center in the downregulation of the immune response induced by flg22, while the NC group showed a relatively moderate effect on PSII activity (Table 1). *Gao et al. (2012)* treated tobacco leaves infected with pathogenic microorganisms with the $Ca^{2+}$ channel inhibitor $LaCl_3$ and obtained results consistent with our own. In the flg22-induced response, $Ca^{2+}$ actively reduces the $PI_{(abs)}$. It seems that $Ca^{2+}$ and chitohexose have differing effects on the PSII reaction center activity as the latter did not significantly affect $PI_{(abs)}$ (Table 1).

CP12 is a chloroplast-localized and photosynthesis-related protein that participates in the Calvin cycle of photosynthesis by interacting with glyceraldehyde 3-phosphate dehydrogenase (GAPDH) and phosphoribulokinase (PRK) (*Rocha & Vothknecht, 2013*). *CP12* expression differed under the flg22 and chitohexose treatments, with chitohexose inhibiting its expression and thus restricting the Calvin cycle and photosynthesis. At the same time, $Ca^{2+}$ participation can still inhibit *CP12* gene expression (Fig. 2).

We examined the expression levels of *PsbS* and *VDE*, which participate in stabilizing PSII. Compared with the effect of flg22, chitohexose induced the downregulation of *PsbS* and the upregulation of *VDE*, with $Ca^{2+}$ playing a more significant role in regulating the expression of *VDE*. *PsbS* expression can improve the photosynthetic capacity and *VDE* expression can protect photosynthetic organs from damage by excess light energy, which indicates that chitohexose can reduce photosynthesis to protect the photosynthetic organs from damage by excess light energy (*Sylak-Glassman et al., 2014*).

NPQ is an important pathway that protects the plant photosynthetic reaction centers. NPQ can be used to reflect the ability of plants to dissipate excess light energy, with more NPQ providing the plant with a stronger photoprotection. NPQ is associated with $Ca^{2+}$ signaling (*Yang et al., 2013*). Here, we showed that flg22 and chitohexose reduced the heat dissipation, while $Ca^{2+}$ further inhibited this process (Figs. 3B and 3D). In addition to NPQ, the reaction center inactivation also helps plant to dissipate excess energy when treated with $Ca^{2+}$, because of the decreased expression of *CP12* (Figs. 2B and 2D). The decrease of heat dissipation may be attributed to the fact that plants can increase their energy utilization rate by reducing energy dissipation, which causes the rapid eruption of ROS (Fig. 4), thereby triggering the immune response (*Göhre et al., 2012*).

Flg22 and chitohexose can enhance the plant immune response by reducing photosynthesis, thus focusing the available resources on enhancing disease resistance. In this study, these treatments inhibited photosynthesis by downregulating *PsbS* and *CP12* expression (Fig. 2) while upregulating *VDE*, and decreasing heat dissipation and the non-photochemical energy dissipation of peanut leaves (Fig. 3). This suggests that flg22 and chitohexose can enhance the immune response using different pathways to increase

the disease resistance of peanuts, with flg22 focused on downregulating the activity of the photosynthetic reaction centers while chitohexose mainly regulates the accumulation of ROS, especially $O_2^-$.

## CONCLUSIONS

Hypersensitivity reaction is a common mechanism of plant immune response. On the surface of plant cells, there are recognition receptors that stimulate the immune response. The conserved components of pathogens that can be recognized by receptors, such as flagellin flg22 and chitosan. To date, many studies have explored the regulation of photosynthesis under stress conditions, but relatively few studies have examined the regulation of photosynthesis by the immune response, and the PAMP-triggered regulation of peanut photosynthesis has not been reported. Here, we found that flg22 and chitohexose triggered the peanut immune pathways through $Ca^{2+}$ signaling, with flg22 decreasing the activity of the photosynthetic reaction center in a $Ca^{2+}$-mediated manner. The absence of $Ca^{2+}$ alleviated the damage to PSII caused by the immune response, but the activity of PSII would be affected. The immune pathway triggered by chitohexose had less influence on photosynthetic electron transport than the pathway triggered by flg22. Both PAMP treatments reduced the energy dissipation, while the downregulation of NPQ required $Ca^{2+}$ participation.

## LIST OF ABBREVIATIONS

| | |
|---|---|
| $CO_2$ | Carbon dioxide |
| CP12 | Chloroplast protein 12 |
| Flg22 | flagellin 22 |
| $H_2O_2$ | Hydrogen peroxide |
| GAPDH | Glyceraldehyde 3-phosphate dehydrogenase |
| MAPK | Mitogen-activated protein kinase |
| NPQ | Non-photochemical quenching |
| PAMP | Pathogen-associated molecular patterns |
| PCR | Polymerase Chain Reaction |
| PEA | Plant efficiency analyser |
| PFD | Photon flux density |
| PR | Pathogenic related gene |
| PRK | Phosphoribulokinase |
| PRR | Pattern recognition receptors |
| Psb | Photosystem b |
| PSI | Photosystem I |
| PSII | Photosystem II |
| ROS | Reactive oxygen species |
| VDE | Violaxanthin de-epoxidase. |

### Funding

This study is supported by the Natural Science Foundation of China (32272020), Natural Science Foundation of Shandong Province (ZR2023MC109), Shandong Key R&D Program (Major Scientific and Technological Innovation Project) (ZFJH202310), Shandong Academy of Agricultural Sciences innovation project (CXGC2023F13), Huangshan science and technology plan project (2022KN-02). The funders had no role in study design, data collection and analysis, decision to publish, or preparation of the manuscript.

### Grant Disclosures

The following grant information was disclosed by the authors:
 Natural Science Foundation: 32272020.
Natural Science Foundation of Shandong Province: ZR2023MC109.
Shandong Key R&D Program (Major Scientific and Technological Innovation Project): ZFJH202310.
Shandong Academy of Agricultural Sciences innovation project :CXGC2023F13.
Huangshan science and technology plan project: 2022KN-02.

### Competing Interests

The authors declare that they have no competing interests.

### Author Contributions

- Quan Wang performed the experiments, analyzed the data, prepared figures and/or tables, authored or reviewed drafts of the article, and approved the final draft.
- Ye Zhang performed the experiments, prepared figures and/or tables, authored or reviewed drafts of the article, and approved the final draft.
- Li Cui performed the experiments, authored or reviewed drafts of the article, performed part of the experiments, and approved the final draft.
- Jingjing Meng performed the experiments, authored or reviewed drafts of the article, performed part of the experiments, and approved the final draft.
- Sha Yang analyzed the data, prepared figures and/or tables, authored or reviewed drafts of the article, and approved the final draft.
- Xinguo Li conceived and designed the experiments, authored or reviewed drafts of the article, and approved the final draft.
- Shubo Wan conceived and designed the experiments, authored or reviewed drafts of the article, and approved the final draft.

### Data Availability

  The raw data is available in the Supplemental File.

## Supplemental Information

Supplemental information for this article can be found online at http://dx.doi.org/10.7717/peerj.16841#supplemental-information.

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
