# Peer review of "Different roles of Ca2+ and chitohexose in peanut (Arachis Hypogaea) photosynthetic responses to PAMP-immunity"

_PeerJ, doi:10.7717/peerj.16841_

## Round 0.1 · original submission · Minor Revisions

Revise the article considering the reviewers' comments.

·

Basic reporting

Clear and unambiguous, professional English used throughout.

Experimental design

Methods described with sufficient detail & information to replicate

Validity of the findings

All underlying data have been provided; they are robust, statistically sound, & controlled.

Additional comments

MS is well written and research finding are interested and novel. However, some minor modification is needed to improve the MS.
1. Aims of the study or hypothesis is not given.
2. Elaborate the result part with some data showing how much fold expression or how much per cent increased or decreased.
3. Remove general statement form discussion and conclusion part (as highlighted in attached pdf).
4. Some minor correction are highlighted in the attached pdf.

Reviewer 2 ·

Basic reporting

The manuscript is interesting and would be useful for peanut researchers. However, there are some lacuna or need clarification in the materials and methods parts which may be rectified. Modification of the manuscript by clarifying or incorporating the suggestions would be highly useful to the readers to understand the article.
Moreover, there are some spelling errors or typo errors in the manuscript. Some of the Examples are listed here.
1. In the manuscript, both FLG22 or flg22 were found which may be uniformly written.
2. Correct the sentence of Line 355 "photosynthesis, thus improving the disease resistance of plants. inhibited photosynthesis"

Experimental design

The materials and methods of the manuscript were not written in proper ways. The following points are suggested for improvement of the manuscript.
1. What is the concentration of Calcium nitrate tetrahydrate used in the Hoagland nutrient solution?
2. What is the condition of light and temperature for incubation with FLG22 and 200 g ml-1 chitohexose for different hours?
3. How much of leaves sample of peanut were taken for treatment with FLG22 and chitohexose?
3. Some of the abbreviations are required to write expanded form. Examples are PEA, NQP, CP12, PsbS, PsbP, etc.
4. Primer sequences of PsbO and PsbP genes were missing in the manuscript/material and methods.
5. In the manuscript, four genes such as PR-4, PR-10, PsbO and PsbP genes where selected for expression analysis through real time PCR. However, the expression analysis of last two genes, PsbO and PsbP genes were not included in the results and discussion parts. This may be rectified.
6. In the result part, FLG22 and chitohexose induced CP12 and PsbS gene expression were mentioned. However, the expression analysis of these two genes, CP12 and PsbS were not given in the materials and method parts. Similarly, expression analysis of the Violaxanthin de-epoxidase (VDE) were not included in the materials and methods.

Validity of the findings

After rectifying the materials and materials and incorporating as suggested in experimental design in the results, the finding of the manuscript would be further improved.

Additional comments

NA

---

## Round 0.2 · Minor Revisions

Your revised article has been approved by reviewers. But the manuscript needs professional editing.  Consider that even the definition of PAMP is incorrect; they are pathogen-associated molecular patterns, not pathology-related molecular patterns (line 31).  The next sentence of the abstract "This immune triggering pattern is relatively conservative and is the result of long term plant evolution" has multiple errors.  The pattern is "relatively conserved" (not conservative).  And then "the result of long term plant evolution" makes no sense.  These are microbial pattern, not plant patterns.  Maybe the authors mean that recognition of these patterns is the result of long term plant evolution but that is not what is said.  I stopped reading at this point, but a quick skim suggests that the whole paper needs editing." Revise the article and while resubmitting submit a certificate from professional editing services.

**Language Note:** The Academic Editor has identified that the English language must be improved. PeerJ can provide language editing services - please contact us at [email protected] for pricing (be sure to provide your manuscript number and title). Alternatively, you should make your own arrangements to improve the language quality and provide details in your response letter. – PeerJ Staff

·

Basic reporting

Well written

Experimental design

Adequate

Validity of the findings

Proper

Additional comments

Paper is revised as per reviewers suggestions thus can be accepted.

Reviewer 2 ·

Basic reporting

The manuscript is revised with the suggestions of the reviewers. It is found clear and suitable for publication.

Experimental design

The experimental method is also improved in the revised manuscript. However, the accession number of the NCBI for PR-4, PR-10, VDE, CP12 genes and photosystem b (Psb) family gene may be included in the materials and methods part.

Validity of the findings

All are well written.

Additional comments

No comments.

---

## Round 0.3 · Minor Revisions

Article may be accepted publication as the language has improved. but still the following issues have been identified

line 34 "such receptors". No receptors have been mentioned yet.

line 35-36 "it is unclear whether photosynthesis is affected by PAMP...". Incorrect. A 2002 paper did studies similar to those described here (but actually in more detail). The authors need to cite previous work and explain what is different/new about their study (or is it just confirmational?)
https://pubmed.ncbi.nlm.nih.gov/22550958/

line 105 "Our previous studies". The paper cited is not by the authors of the current manuscript."

Revise the article accordingly.

---

## Round 0.4 · accepted · Accept

As improved considering the previous suggestions, article accepted for publication.

> line 105 "our previous studies" should be changed to "our previous studies and those of others" since one of the two papers cited is not by these authors.